# Obesity and Pancreatic Cancer: Recent Progress in Epidemiology, Mechanisms and Bariatric Surgery

**DOI:** 10.3390/biomedicines10061284

**Published:** 2022-05-31

**Authors:** Shuhei Shinoda, Naohiko Nakamura, Brett Roach, David A. Bernlohr, Sayeed Ikramuddin, Masato Yamamoto

**Affiliations:** 1Department of Surgery, University of Minnesota, Minneapolis, MN 55455, USA; shino065@umn.edu (S.S.); nnakamur@umn.edu (N.N.); roach229@umn.edu (B.R.); ikram001@umn.edu (S.I.); 2Department of Biochemistry, Molecular Biology and Biophysics, University of Minnesota, Minneapolis, MN 55455, USA; bernl001@umn.edu; 3Masonic Cancer Center, University of Minnesota, Minneapolis, MN 55455, USA

**Keywords:** PDAC, pancreatic cancer, obesity, bariatric surgery

## Abstract

More than 30% of people in the United States (US) are classified as obese, and over 50% are considered significantly overweight. Importantly, obesity is a risk factor not only for the development of metabolic syndrome but also for many cancers, including pancreatic ductal adenocarcinoma (PDAC). PDAC is the third leading cause of cancer-related death, and 5-year survival of PDAC remains around 9% in the U.S. Obesity is a known risk factor for PDAC. Metabolic control and bariatric surgery, which is an effective treatment for severe obesity and allows massive weight loss, have been shown to reduce the risk of PDAC. It is therefore clear that elucidating the connection between obesity and PDAC is important for the identification of a novel marker and/or intervention point for obesity-related PDAC risk. In this review, we discussed recent progress in obesity-related PDAC in epidemiology, mechanisms, and potential cancer prevention effects of interventions, including bariatric surgery with preclinical and clinical studies.

## 1. Introduction

The prevalence of obesity is increasing in the United States (U.S.) and other countries. The overall prevalence of obesity based on physical measurements data in the U.S. was 41.5% and that of severe obesity (BMI > 35) was 20.7% [1]. Obesity is a major risk factor for the development of metabolic syndromes and many cancers, translating to a decrease in longevity. In particular, it has been known that obesity raises the risk of obesity-related malignancy, including esophageal adenocarcinoma, endometrial, gastric, liver, renal, colorectal, breast, ovarian, and pancreatic cancers [2,3,4]. Pancreatic ductal adenocarcinoma (PDAC) is the third leading cause of cancer-related death, and 5-year survival of PDAC remains at approximately 9% in the U.S. [2]. Obesity is a known risk factor for PDAC, as well as family history, type 2 diabetes, and tobacco use [2]. Bariatric surgery, which is an effective treatment for severe obesity with 25–30% weight loss, has been shown to reduce the risk of PDAC [3,5]. Furthermore, the involvement of visceral adipose tissue (VAT) and fatty infiltration of the pancreas in pancreatic carcinogenesis has been suggested [6,7]. Interestingly, a multicenter retrospective study of early pancreatic cancer in Japan demonstrated that focal fat replacement was detected in 42.0% of Stage 0 and 41.8% of Stage 1 cases [8], suggesting that focal parenchymal atrophy and fat replacement may be specific to early-stage PDAC. A better understanding of the link between obesity and PDAC is important for the identification of a novel marker and/or therapeutic targets for obesity-related PDAC risk. The scope of this review includes recent progress in obesity-related PDAC in epidemiology, the biological mechanisms, and potential cancer prevention effects of interventions, including bariatric surgery with preclinical and clinical studies (Figure 1).

## 2. Epidemiology of Obesity and PDAC

The incidence of obesity has increased over the past decades in many countries [9]. In the U.S., the age-adjusted prevalence of obesity in 2017–2018 was 40.0% among younger adults aged 20–39, 44.8% among middle-aged adults aged 40–59, and 42.8% among older adults aged 60 and over [10]. Along with the increased prevalence of obesity, the burden of cancers related to excess body weight has increased worldwide from 1990 to 2019 [11]. Similarly, large epidemiology studies indicate the increased risk of pancreatic cancer in obese patients [12,13]. In a recent report based on the Global Cancer Observatory and the Global Health Observatory databases in 2020, PDAC prevalence showed positive correlations with obesity (r = 0.508, *p* < 0.001), insufficient physical activity (r = 0.417, *p* < 0.001), and high cholesterol (r = 0.780, *p* < 0.001) [13]. In the multivariate analysis, high cholesterol and obesity were identified as positive predictors for the incidence of PDAC [14]. We reviewed articles from the last five years indicating the risk of PDAC in obese patients as shown in Figure 2 (excluding review articles) [14,15,16,17,18,19,20,21,22,23,24,25]. These studies were conducted using a large-scale biomedical database that comprises demographic, clinical, biochemical, and genetic data. Overweight and obesity were defined by BMI, the ratio of weight in kilograms (kg) to height in meters squared (m^2^), where a BMI of 18.5–25 is normal, 25–30 is overweight, and ≥30 is obese. Among the recent reports, obesity (BMI ≥ 30) was associated with a higher risk of PDAC incidence with a hazard ratio (HR) between 1.12 and 1.71 [14,15,16,17,18]. In addition, elevated BMI by 4.6 kg/m^2^ increased the HR to 1.34 [19]. Overweight (BMI, 25 to 30 or 85th to 95th percentile), waist circumference (≥95 cm in females), and metabolic syndrome conditions also increased the risk of incidence [21,22,23,24]. These results were compatible with data from a recent meta-analysis detailing the relationship between obesity and PDAC risk; the relative risk for a 5 kg/m^2^ increment in BMI was 1.10 (95% CI 1.07–1.14) and for a 10-cm increase in waist circumference was 1.11 (95% CI 1.05–1.18), with no significant heterogeneity [26]. As for the differences between males and females, a previous report found that obesity was a risk factor among women but not men [27]. However, the meta-analysis concluded that the relative risk of obesity in PDAC was similar among men and women [22]. Specifically noted is that the body weight change from age 21 was reported to be associated with an increased risk of PDAC (body weight HR per 10 lbs, 1.03; 95% CI, 1.01–1.05 and BMI HR per 5 kg/m^2^, 1.08; 95% CI, 1.01–1.15) [28]. It was also shown that the HR caused by a 5 kg/m^2^ increment in BMI declined steadily from 1.25 in persons aged 30–49 years to 1.13 in those aged 70–89 years [29]. In addition, both men and women who were obese as adolescents had an increased risk of subsequent PDAC [22]. Taken together, these results suggest that BMI before 50 years of age is more strongly associated with PDAC risk than BMI at older ages.

Regarding the mortality of PDAC with obesity, Huang et al. analyzed mortality using the GLOBOCAN database in 184 countries from 2003 to 2018 [30]. They showed that countries with higher education, income, and life expectancy tended to have higher mortality of PDAC as age-standardized rates. The countries with higher mortality of PDAC were more likely to have a higher prevalence of smoking, alcohol drinking, physical inactivity, obesity, hypertension, and high cholesterol. As mentioned before, obesity is strongly associated with an increased risk of PDAC incidence, but it still remains unclear whether obesity could increase mortality in PDAC patients. A systematic review in 2016 evaluated an increase in PDAC-related mortality among overweight (BMI, 25 to <30; HR, 1.06; 95% CI 1.02–1.11) and obese (BMI ≥ 30; HR, 1.31; 95% CI 1.20–1.42) individuals [31]. This might be affected by surgical difficulty or resection status in curative treatment for obese PDAC patients because obese PDAC patients tended to show poor long-term prognosis following surgery [32]. Interestingly, increasing levels of obesity were associated with increased mortality in Western populations (HR, 1.32; 95% CI, 1.22–1.42) but not in Asian-Pacific populations (HR, 0.98; 95% CI, 0.76–1.37). In both East Asians and South Asians, obesity was unrelated to the mortality of PDAC [33]. Among African Americans who have a higher prevalence of obesity, the HRs in PDAC mortality were reported to be 1.25 and 1.31 for BMI 30 to 34 and BMI ≥ 35, respectively [34]. Therefore, countries, regions, and races should be taken into consideration to assess the relationship between obesity and PDAC mortality. On the other hand, we hypothesize that therapeutic effects could directly contribute to the prognosis in patients with PDAC. In vivo experiments demonstrated that obesity was associated with increased desmoplasia and reduced response to chemotherapy in PDAC [35]. In contrast, a poorer response to chemotherapy for PDAC was reported in sarcopenic patients [36]. Thus, sarcopenic obesity is an important consideration in interpreting mortality data in PDAC in obese patients. Sarcopenic obesity is characterized by the loss of muscle mass accompanied by increased fat mass in patients with aging or malignancy and is known to be an independent predictor of survival in patients with solid tumors in the respiratory and gastrointestinal tract [37]. In a previous meta-analysis, sarcopenic obesity was reported in 0.6% to 25.0% of PDAC patients and was significantly associated with lower overall survival (HR, 2.01; 95% CI, 1.55–2.61) [38]. Sarcopenic obesity was also an independent predictor of major postoperative complications in PDAC patients who underwent curative resection [39]. Although it is unclear that sarcopenic obesity could affect the response to chemotherapy or prognosis after surgery for PDAC, sarcopenic obesity may lead to worse outcomes during the treatment for PDAC and result in impacting prognosis. Further work is needed to provide deeper insight into the relationship between obesity and mortality in patients with PDAC.

## 3. Adipose Tissue, Adipocyte, and Inflammation

### 3.1. Adipose Tissue and Inflammation

Adipose tissue is known to store energy and regulate paracrine and endocrine metabolism. The majority of adipose tissue is white adipose tissue (WAT), which is primarily comprised of large adipocytes that harbor a single lipid droplet and markedly fewer mitochondria than brown adipocytes [40]. WAT is distributed throughout the body in several distinct regions. These include visceral adipose tissue (VAT), which includes omental and mesenteric WAT. Obesity is associated with the accumulation of pro-inflammatory cells in VAT, and chronic low-grade unresolved inflammation is a typical characteristic of obesity [41]. Hypoxia that results from restricted blood vessel growth and decreased blood flow due to the rapidly expanding adipose tissue is linked to adipocyte dysfunction and immune cell recruitment. Diet-induced obesity is associated with the development of type 1 inflammatory responses in VAT, characterized by IFN-γ [42]. NK cells and CD8(+) T cells in adipose tissue produce IFN-γ, driving Adipose tissue macrophage polarization of M1 macrophages. This switch of anti-inflammatory M2 macrophages to a pro-inflammatory M1-like phenotype is considered an important step in the early phase of obesity-induced adipose tissue inflammation [42]. Consequently, a rapid accumulation of pro-inflammatory cells in VAT follows, leading to systemic pro-inflammatory cytokines and chemokines, including IL-6, TNF-a, IL-1b, IL-18, and MCP-1 [42]. These elevated factors in VAT also stimulate neoplastic pancreatic epithelial cells, cancer-associated fibroblasts, and immune cells. Although BMI is widely used as a marker for general adiposity, meta-analysis has identified that VAT volume has a stronger correlation to certain gastrointestinal malignancies, including PDAC [26], given that the pancreas is anatomically surrounded by VAT [35]. Recently, lipid deposition in the pancreas (fatty pancreas) has been studied regarding the relationship with non-alcoholic fatty pancreatic diseases (NAFPD) [43,44], and the involvement of fatty pancreas in pancreatic carcinogenesis has been suggested [6,7]. Interestingly, some groups have demonstrated that cancer-associated adipocytes (CAAs), as well as infiltrating inflammatory and immune cells in the peripancreatic adipose tissue, can regulate tumor cells by providing adipokines, proinflammatory cytokines, chemokines, and growth factors, which may accelerate pancreatic neoplasia [45].

### 3.2. Adipokines

Adipokines promote the growth, migration, and invasion of cancer cells through oncogenic signaling or indirect mechanisms, such as insulin resistance, angiogenesis, and regulation of the immune response [46]. Leptin and adiponectin are common adipokines, and obesity increases leptin levels while decreasing adiponectin levels [47]. The link between leptin and PDAC is still unclear, but a prospective case-control study reports elevated levels of leptin as a PDAC risk in men [48]. In vivo studies demonstrate that leptin can promote PDAC proliferation [49]. Furthermore, knockdown of the leptin receptor in pancreatic cancer cells leads to functional impairment of leptin signaling and inhibits orthotopic tumor growth in obese mice [49].

Adiponectin is a well-described adipokine secreted by adipocytes. Multiple studies have shown the protective role of adiponectin against obesity-associated diseases and cancer. Lower levels of adiponectin are known to be associated with an increased risk of PDAC in the prospective study [48]. This prospective study also suggested the underlying mechanism may regulate glucose metabolism and insulin resistance. Messaggio F., et al. reported that adiponectin attenuates the risk of PDAC through the activation of the adiponectin receptor, which inhibits leptin-mediated STAT3 activation using in vitro and in vivo studies [50]. Jiang J., et al. reported that adiponectin decreases PDAC risk through the inactivation of GSK-3β (glycogen synthase kinase 3 beta), an enzyme that plays an important role in the WNT signaling pathway [51]. 

An in vivo study using the PDAC mice model with diet-induced obesity found that adipokine lipocalin-2(LCN2) expression in PDAC tissue was upregulated [52]. LCN2, which is known to increase with obesity, is positively correlated with inflammation and insulin resistance. Furthermore, the downregulation of LCN2 leads to reduced inflammation, fibrosis, and the incidence of pancreatic intraepithelial neoplasia (PanIN), a microscopic neoplastic lesion of the pancreas that can progress to invasive ductal adenocarcinoma [52]. These studies revealed the important role of adipokines in linking obesity and PDAC.

### 3.3. Cytokines, Chemokines, and Senescence-Associated Secretory Phenotype (SASP)

Cytokines, chemokines, and their receptors influence a variety of biological functions, including inflammatory responses, immune-cell trafficking, angiogenesis, and metastasis [53]. Incio J., et al. reported that obesity increased IL-1β levels and immune cell infiltration in PDAC, which was associated with increased tumor growth and metastasis [54]. Philip B., et al. reported that high fat diet-induced inflammation can activate oncogenic KRAS and its downstream targets (COX2 and phospho-ERK) to promote the activation of pancreatic stellate cells (PSCs). Furthermore, there is an increase in infiltration of macrophages, leading to the establishment of a positive feed-forward loop to maintain KRAS activity, which further increases inflammation. These changes result in increased fibrotic stroma, more PanINs and PDACs, and shortened survival in a transgene mouse model [55]. Dawson D.W., et al. reported that the pancreas of a conditional KrasG12D mouse model fed a high-fat diet shows obvious signs of inflammation, increased numbers of infiltrating inflammatory cells, increased levels of cytokines and chemokines, increased interstitial fibrosis, and aggravation of PanIN lesions [56]. Zhang T., et al. demonstrated that chemokines CXCL1 and CXCL8 chemoattract adipose stromal cells by signaling through their receptors, CXCR1 and CXCR2, in cell culture models [57]. This work sheds light on how cytokines and chemokines secreted by adipose tissue affect the pancreas. 

Cellular senescence is characterized by irreversible cell cycle arrest triggered by DNA damage, telomere attrition, oxidative damage, mitotic stress, mitochondrial dysfunction, and endoplasmic reticulum stress as well as by oncogene activation [58,59]. The accumulation of senescent cells potentiates aging, decreases the health span, and plays a causative role in age-related diseases, including cancer [59,60]. A variety of adipose cells are reported to become senescent, including preadipocytes, adipocytes, and immune cells [61,62]. 

Cellular senescence is controlled by tumor suppressor genes and seems to be a checkpoint that prevents the growth of cells at risk for carcinogenesis [63,64]. However, the effects of senescent cells within the tumor microenvironment are complex [60,65,66]. Senescent cells synthesize and secrete a specialized set of signaling molecules that affect the microenvironment of adipose and surrounding tissues [61]. These factors, generally referred to as the senescence-associated secretory phenotype (SASP), include cytokines, chemokines, extracellular matrix proteases, growth factors, extracellular vesicles, and other signaling molecules [61,67,68]. Krtolica et al. reported that senescent fibroblasts promote epithelial cell growth and tumorigenesis [69]. Furthermore, chronic inflammation by senescence-associated secreted factors, such as IL-22, might be contributors to PDAC [60,70,71]. Future work might reveal how SASP regulates the tumor microenvironment.

## 4. Insulin, Insulin-like Growth Factor, and Insulin-like Growth Factor-I Receptor

The prevalence of diabetes among patients with PDAC is high. In a study of 512 PDAC patients and 933 control subjects, diabetes was found in 47% of PDAC patients compared with only 7% of control subjects [72]. On the other hand, PDAC is also a cause of pancreatogenic diabetes, a form of secondary diabetes classified by the American Diabetes Association and the World Health Organization as type 3c diabetes mellitus (T3cDM) [73]. In 74% of PDAC patients with diabetes, the diagnosis of diabetes was made within 24 months before the diagnosis of PDAC [74]. Interestingly, in an effort to find biomarkers capable of stratifying new-onset type 2 diabetes subjects into those with type 2 diabetes mellitus (T2DM) and those with PDAC-related diabetes, Oldfield L., et al. showed that the combination of adiponectin and interleukin-1 receptor antagonist (IL-1Ra) has strong diagnostic potential (AUC of 0.91; 95% CI: 0.84–0.99) for the distinction of PDAC-related diabetes from T2DM [75]. The mechanism behind PDAC causing T3cDM has not been fully elucidated, but it is clear that patients with long-standing T2DM, often associated with obesity, have an increased risk of PDAC [73]. Obesity is associated with increased basal and post-prandial insulin secretion [76,77,78]. High levels of insulin activate the IGF receptor, thereby potentiating the mitogenic and tumor-promoting effects [73]. KC mice subjected to diet-induced obesity consistently developed hyperinsulinemia and elevated levels of IGF-1. Furthermore, metformin significantly suppressed PDAC development in KC mice with diet-induced obesity, which was associated with the normalization of hyperinsulinemia [79]. IGF-1 is a peptide hormone, structurally similar to insulin. The binding of insulin or IGF-1 to the insulin receptor or to the insulin-like growth factor-I receptor (IFG-1R) is known to inhibit lipolysis [80], and consequently, insulin resistance is correlated with increased adipocyte lipolysis. Furthermore, the binding of IGF-1 to IGF-1R initiates downstream signals that activate PI3K/Akt/mTOR and MEK/Erk pathways, which stimulate cancer cell proliferation and induce drug resistance [81]. Prospective studies for patients with metastatic cancer reveal that gemcitabine plus the IGF-1R antagonist (MK-0646) improved OS compared with gemcitabine plus erlotinib even though tissue and serum IGF-1 did not correlate with the clinical outcome [82]. Further research is needed to clarify the impact of IGF-1R.

## 5. Lipid Metabolism

### 5.1. Dyslipidemia (Triglycerides and Cholesterol)

Obesity is associated with dyslipidemia, which is reported to be mainly driven by the effects of insulin resistance and pro-inflammatory adipokines [83]. Metabolic dyslipidemia, arising from insulin resistance and obesity, leads to high concentrations of triglycerides (TG) accompanied by decreased high-density lipoprotein cholesterol (HDL-C) concentrations [83]. Insulin suppresses lipolysis in adipose tissue and controls the release of FFAs into circulation [84]. Moreover, insulin stimulates apolipoprotein B-100 (apoB-100) degradation and suppresses very-low-density lipoprotein (VLDL) secretion from the liver [85]. In the liver, insulin dephosphorylates and activates 3-hydroxy-3-methylglutaryl-CoA (HMG-CoA) reductase, leading to the stimulation of cholesterol synthesis [84]. In the state of insulin resistance, plasma clearance of TG-rich lipoproteins (chylomicron and VLDL) is delayed, resulting in hypertriglyceridemia. Excess triglyceride levels in obesity lead to adipocyte hypertrophy cell hypoxia and macrophage infiltration into pancreatic tissue [86].

In PDAC cells, cholesterol acquisition highly relies on uptake from outside. Compared with the modestly increased cholesterol synthesis pathway, strong activation of low-density lipoprotein receptor (LDLR)-mediated uptake of cholesterol-rich lipoproteins is observed in murine PDAC cells [87]. High levels of total cholesterol and triglycerides, or low levels of high-density lipoprotein and apolipoprotein A-I, have been identified to be associated with an increased risk of obesity-related cancers, including PDAC, in a meta-analysis [88]. On the other hand, another meta-analysis demonstrated that total serum cholesterol is inversely correlated with PDAC risk in men [89]. Statins, HMG-CoA reductase inhibitors, are a class of lipid-lowering medications that reduce the risk of cardiovascular disease [90], and some groups reported that statins reduce PDAC risk and improve survival in patients with PDAC [91]. However, a large meta-analysis showed no statistically significant associations between statin use and cancer risk [92]. A phase II clinical trial combining simvastatin with gemcitabine in advanced PDAC treatment failed to show a clinical benefit (NCT00944463) [93]. Further studies are therefore needed to clarify the connection between dyslipidemia and PDAC.

### 5.2. Fatty Acids

Dysregulation in lipid metabolism is one of the important metabolic changes in cancer. Lipogenic activity is activated in PDAC cells to obtain energy for proliferation, survival, invasion, metastasis, and response to cancer therapy [94]. In contrast to normal cells relying on dietary fat, approximately 93% of triacylglycerol fatty acids in tumor cells are de novo synthesized from mitochondrial citrate [95,96]. Carbon from glucose and glutamine contribute to citrate production [97]. Among different fatty acids, saturated and monounsaturated fatty acids are considered to promote the growth of PDAC cells [98]. Polyunsaturated fatty acids, mainly containing the omega-3 and omega-6 families, have opposing effects on PDAC. Omega-3 fatty acids inhibit cancer cell proliferation via reducing AKT phosphorylation, but omega-6 fatty acids increase AKT phosphorylation [99]. However, the results of clinical studies do not suggest a consistent association. A transcriptomics and metabolomics study revealed that lipase and a panel of fatty acids are significantly decreased in pancreatic tumors, and two saturated fatty acids, palmitate and stearate, showed an obvious ability to inhibit the proliferation of PDAC cells [100]. The role of fatty acids in PDAC is complex, and other factors modulating fatty acids may have to be taken into consideration.

### 5.3. Fatty Acid Binding Protein (FABP)

Fatty acid trafficking in cells is known to affect many aspects of cellular function. Fatty acid-binding proteins (FABPs), a family of intracellular lipid chaperones, regulate lipid trafficking and responses in cells [101]. FABPs are small water-soluble proteins that can affect lipid fluxes, metabolism, and signaling within cells. Twelve FABP family members have been identified. Considering where FABPs are expressed, FABP4 and 5 might be related to PDAC growth (Table 1) [102,103,104,105,106,107,108,109]. Furthermore, the role of abnormal FABP expression and function has been implicated as a potential mediator of tumorigenesis [110].

Fatty acid binding protein 4 (FABP4; alternatively called adipocyte protein 2, aP2), which is mainly expressed in adipocytes and macrophages, plays an important role in the development of insulin resistance and atherosclerosis in relation to metabolically driven chronic inflammation [101]. Circulating FABP4 levels are also known to be associated with several aspects of metabolic syndrome [101]. 

FABP4 has also been linked to the development and progression of a variety of cancers [111], where transcriptional regulation downstream of FABP4 is associated with cellular redox status [112]. We reported that the transcriptional activity of nuclear factor E2-related factor 2 (NRF2) was induced in response to the extracellular addition of FABP4, and FABP4 treatment also led to the downregulation of reactive oxygen species (ROS) activity in PDAC cells [113]. Additionally, FABP4 is robustly expressed in macrophages and indirectly promotes cancer progression by altering matrix metalloproteinase (MMP) activity and promoting IL6 [114]. In patients with PDAC, higher FABP4 expression in the tumors measured by immunohistochemistry was correlated with both disease progression and poor prognosis [115]. 

Fatty acid binding protein 5 (FABP5; alternatively called epidermal-FABP, cutaneous fatty-acid-binding protein) is abundantly expressed in skin epidermis, adipocytes, macrophages, and dendritic cells [116]. FABP5 delivers both saturated and unsaturated long-chain fatty acids from the cytosol to nuclear peroxisome proliferator-activated receptors (PPARs), resulting in increased proliferation, carcinogenesis, and metastasis [111,116]. The inhibition of FABP5 in prostate cancer cell lines with its antisense RNA decreased VEGF and micro vessel densities in tumors [117]. Decreased tumor growth and lung metastasis were observed in FABP5-/- mice orthotopically injected with murine breast cancer cells, and clinical FABP5 RNA levels correlated with EGFR expression and poor prognosis [118]. Hughes, et al. demonstrate that there is significantly lower FABP5 expression in activated fibroblasts or PSCs in PDAC compared to quiescent fibroblasts [119]. FABPs (particularly FABP4 and 5) may thus become good targets for PDAC therapy and other metabolic diseases.

## 6. Tumor Microenvironment (TME)

The importance of the tumor microenvironment (TME) in the development, growth, and progression of cancer is now widely recognized. It is interesting to note that PDAC arises adjacent to adipose depots. This suggests adipocytes can affect oncogenic processes of PDAC via paracrine effects [120]. Furthermore, PDAC cells are surrounded by desmoplasia composed of collagens, resulting in hypoxia and nutrient-poor conditions [121]. In such hypoxic conditions, cancer-associated adipocytes act as a source of fatty acids and lipids for PDAC cells, which contribute to disease progression [122]. Fatty acids also have other aspects as essential mediators of cancer progression and metastasis through the remodeling of the TME [123]. The co-culture of adipocytes and PDAC cells in vitro increases the expression of WNT5a, which converts adipocytes to fibroblast-like cells [124]. Adipocytes influence the immunological landscape in the TME. A high-fat diet (HFD) increases fat uptake in tumor cells but not in tumor-infiltrating cytotoxic T lymphocytes (CTLs). These differential adaptations lead to altered fatty acid distribution in HFD tumors, impairing CD8+ T cell infiltration and function [125]. In a transgenic mouse model of PDAC, obesity in mice induces steatosis and a fibroinflammatory TME in which IL-1B is released by adipocytes. These events induce the recruitment of tumor-associated neutrophils (TANs), activation of PSCs, and aggravation of desmoplasia [35]. Furthermore, PSCs also secrete IL1β, and the inactivation of PSCs reduces IL1β expression and TAN recruitment [35].

## 7. Bariatric Surgery and PDAC

As described above, obesity leads to the development of PDAC, and the prevention of obesity is therefore a good healthcare practice. Currently, bariatric surgery is the most effective and dependable intervention for obese patients. The validity and benefits of bariatric surgery for weight loss and long-term weight maintenance were already indicated in many previous reports [126,127,128,129,130,131]. Bariatric surgery is generally performed in patients with a BMI > 40 or those with BMI > 35 kg/m^2^ with obesity-related comorbidities, who do not achieve weight loss despite a medically prescribed program of diet and exercise [132,133]. There are a number of surgical procedures, such as laparoscopic sleeve gastrectomy and Roux-en-Y gastric bypass (RYGB) [5], and it is estimated that more than 228,000 procedures are performed on annual basis in the U.S. [134]. Among these surgical procedures, laparoscopic sleeve gastrectomy has been widely accepted as the most popular procedure in the U.S. due to its technical simplicity and low complication rate [135,136]. In previous analyses indicating the efficacy of bariatric surgery, a variety of surgical procedures have been included. We thus may need to consider the differences in surgical procedures to determine the long-term benefits in not only weight maintenance but also control of comorbidities. Effective weight loss by bariatric surgery contributes to the improvement of obesity-related comorbidities as well as cancer incidence in obese patients. Previously, bariatric surgery was reported to reduce cancer incidence in some cancers, including breast, endometrial, colorectal, melanoma, and non-Hodgkin lymphoma [137,138]. A Swedish obese subjects’ study, involving 2010 obese patients who underwent bariatric surgery and 2037 contemporaneously matched obese controls, showed that patients in the bariatric surgery group had a hazard ratio (HR) for overall incident cancer of 0.67 (95% CI 0.53–0.86; *p* = 0.0009) compared with patients in the control group [137]. A similar retrospective study was conducted among 9949 patients who underwent bariatric surgery in the U.S. and indicated that cancer-specific mortality decreased to 60% in patients who underwent bariatric surgery for a 7-year follow-up period [138]. 

Regarding the relationship between PDAC and bariatric surgery, Rustgi, et al. conducted a retrospective cohort study using Marketscan Data of newly diagnosed nonalcoholic fatty liver disease (NAFLD) patients with severe obesity between 2007 and 2017 [3]. In total, 98,090 patients were included in the study, and 33,435 (34.1%) received bariatric surgery. This study revealed that patients who received bariatric surgery had a significantly lower risk of PDAC incidence (HR 0.46; 95% CI 0.21–0.93). Schauer, et al. also conducted a retrospective study in a large multisite cohort using 22,198 subjects who had bariatric surgery and 66,427 nonsurgical subjects matched on sex, age, study site, body mass index, and Elixhauser comorbidity index [139]. This study demonstrated that patients undergoing bariatric surgery had a significantly lower risk of obesity-associated cancer incidence (HR 0.59; 95% CI 0.51–0.69; *p* < 0.0001). Interestingly, the risk of PDAC was much lower in patients who had undergone bariatric surgery compared with matched nonsurgical patients (HR, 0.46; 95%, CI 0.22–0.97; *p* = 0.04). A Canadian study also reported that 0.1% of patients with bariatric surgery developed PDAC, compared to 0.33% in the control group, but there was no significant difference between the groups (*p* = 0.166) [140]. Data from the Utah Cancer Registry evaluated that bariatric surgery reduced the incidence of obesity-associated cancers but not PDAC (HR 1.75; 95% CI 0.66–4.63; *p* = 0.26), compared with the obese control group [141]. In this study, all the patients (n = 6,709) in the surgery group underwent RYGB between 1984 and 2002, and the mean follow-up period was 12.5 years. The most recent study by Schauer, et al. that showed the lower HR of PDAC incidence after bariatric surgery included patients undergoing sleeve gastrectomy and was conducted between 2005 and 2012 with a mean follow-up of 3.5 years [139]. Although the reason for the discrepancy in HR between the studies is unclear, the surgical procedures and follow-up period might affect the results. 

It is generally accepted that bariatric surgery reproducibly results in durable weight loss and an improvement in metabolic dysfunction, systemic inflammation, and the adipokine profile [5]. Because of the heterogeneity and follow-up periods, clinical studies are limited in seeking the suppressive mechanism that is involved in cancer development after bariatric surgery. He et al. assessed the effectiveness of bariatric surgery using the genetically engineered Ngn3-Tsc1−/− mouse model that can develop pancreatic cell acinar carcinoma and metastases [142]. This study showed that RYGB reduced the rate of cancer development and improved the survival rate in the mice model. However, they used the mouse model of acinar carcinoma but not PDAC. As far as we know, there is no research that has evaluated the mechanism of bariatric surgery in PDAC development using in vivo experiments. In this context, the establishment of the appropriate PDAC models that can accept interventions of bariatric surgery is essential to evaluate the multiple factors that could be involved in the antitumor mechanism of bariatric surgery. We have already established a mice vertical sleeve gastrectomy model [143]. It might be valuable for PDAC and obesity research to use our surgical method combined with PDAC carcinogenic mice. In addition, Zhou et al. reported antidiabetic effects and the improvement of pancreatic β-cell function in a rat model with gastric bypass surgery [144]. Numerous factors that are involved in insulin resistance, systemic inflammation, lipid metabolism, and gut microbiome could complexly interact with each other and, finally, lead to anti-carcinogenic effects in bariatric surgery. Understanding the dramatic changes of these factors impacted by bariatric surgery can help to decide the role of obesity in carcinogenesis as well as therapeutic targets in PDAC. Therefore, further research is required to clarify the association between PDAC, obesity, and bariatric surgery.

## 8. Conclusions

In this review paper, we have highlighted the recent studies linking obesity and PDAC in epidemiology, mechanisms, and potential cancer prevention effects of interventions, including bariatric surgery. Recent clinical studies suggest that bariatric surgery has also been associated with a protective effect against PDAC development, progression, and mortality. In addition, recent studies of the relationships between obesity and PDAC have indicated many potential mechanisms through which bariatric surgeries realize their clinical benefits shown in outcome studies. Therefore, mechanistic analysis has the possibility for the identification of novel markers and intervention points for obesity-related PDAC. We strongly believe that this approach could improve the current dismal survival rate of patients with PDAC.

## Figures and Tables

**Figure 1 biomedicines-10-01284-f001:**
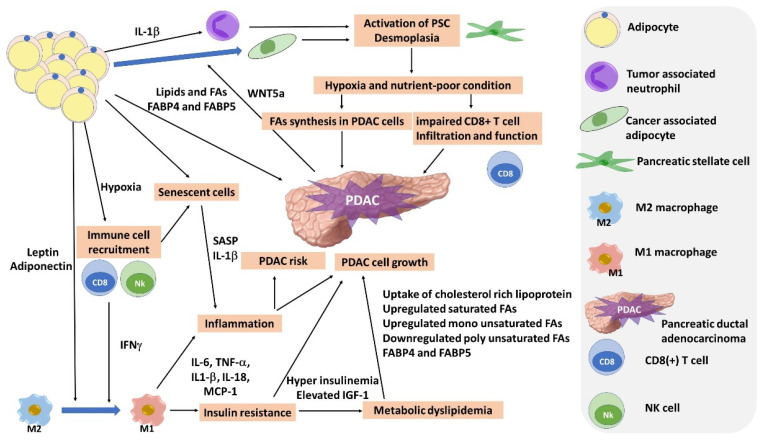
The pancreatic cancer environment in obese individuals. Adipose tissue and its inflammation play a central role in carcinogenesis and the tumor microenvironment. Adipokines, cytokines and chemokine, and other inflammatory mediators secreted by adipose tissue affect the pancreas. Adipose tissue inflammation also causes insulin resistance. Elevated insulin levels are known to cause PDAC cell growth and metabolic dyslipidemia. Dysregulation of lipid metabolism also plays an important role in cancer cells. In particular, (i) enhanced extracellular uptake of cholesterol, (ii) elevated fatty acid synthesis, (iii) imbalance between unsaturated and saturated fatty acids, and (iv) abnormal FABP expressions are important metabolic changes in PDAC cells.

**Figure 2 biomedicines-10-01284-f002:**
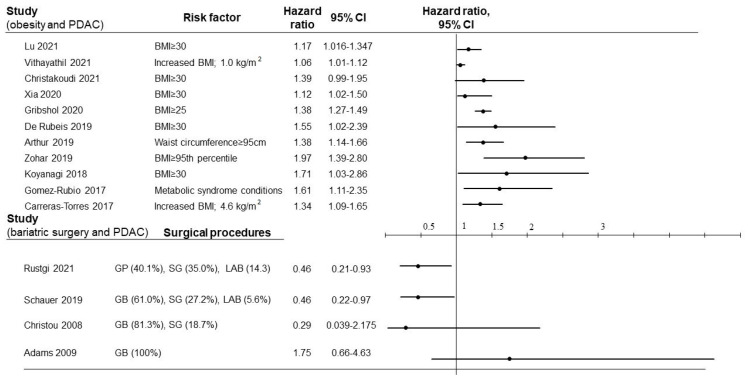
Summary of hazard ratio (HR) for pancreatic ductal adenocarcinoma (PDAC) incidence associated with obesity (above) and bariatric surgery (below). For each plot, the black circle represents the study-specific HR, and the arms of each symbol denote the 95% confidence intervals (95% CI). GB: Gastric bypass, SG: Sleeve gastrectomy, LAB: Laparoscopic adjustable band.

**Table 1 biomedicines-10-01284-t001:** FABP family members and their distribution.

Gene Name	Common Names	Expression	Ligand	Ref
FABP1	Liver FABP	Liver, intestine, pancreas, kidney, lung and stomach	Long-chain FAs, acyl-CoAs and heme	[102,104,105,107]
FABP2	Intestinal FABP	Intestine and Liver	Long-chain FAs	[103,104,105]
FABP3	Heart FABP	Heart, skeletal muscle, brain and many other organs	Long-chain FAs	[104,105]
FABP4	Adipocyte FABP	Adipocyte, macrophage and dendritic cell	Long-chain FAs	[104,105,109]
FABP5	Epidermal FABP	Skin, adipocyte, macrophage, dendritic cell and many organs	Long-chain FAs	[104,105]
FABP6	Ileal FABP	Ileum	Bile acids	[104,105]
FABP7	Brain FABP	Brain	Long-chain FAs and docosahexaenoic acid	[105]
FABP8	Myelin FABP	Peripheral nervous system and schwann cell	Long-chain FAs	[105]
FABP9	Testis FABP	Testis	Long-chain FAs	[105,106,108]
FABP10		Not identified in mammalian species	Long-chain FAs	[106,107]
FABP11		restricted to fishes	Long-chain FAs	[107]
FABP12		human retinoblastoma cell and prostatic cancer cell	Long-chain FAs	[107]

## Data Availability

Not applicable.

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
