# Peer review of "Obesity and Pancreatic Cancer: Recent Progress in Epidemiology, Mechanisms and Bariatric Surgery"

_biomedicines, 2022, doi:10.3390/biomedicines10061284_

Round 1

Reviewer 1 Report

This is an extensive review on the effect of obesity and the metabolic syndrome with its attendant changes in cell signaling and metabolism on the development of pancreas cancer. The authors also review the literature on bariatric surgery and its effect on the incidence of cancer postoperatively.

This review paper does need to be thoroughly edited for proper grammar and word choice.

Author Response

Responses to Reviewer 1’s comments

Thank you very much for carefully reviewing our manuscript and providing valuable comments to us. We have carefully studied the comments and have performed some experiments. We have added some more data and amended the manuscript based on them. We hope our revised manuscript meets with your approval.

Issue: This review paper does need to be thoroughly edited for proper grammar and word choice.

Answer: We asked native English speaker (Brett Roach) to check the grammar and word choice.

Reviewer 2 Report

Comments to authors:

The review demonstrates the relevance of obesity and PDAC with recent studies on epidemiology and highlights mechanisms about how obesity involves or influences the progress of PDAC. Meantime, the review also introduces several recent studies on bariatric surgery, a potential PDAC prevention intervention.

The findings are reasonably discussed and support the conclusions. However, the following points need to be addressed.

Major Points

  1. In the introduction, page 1, lines 29-31. It would be more persuasive if articles or data arecited to prove the conclusion “obesity raises the risk of obesity related malignancy, including esophageal adenocarcinoma, endometrial, gastric, liver, renal, colorectal, breast, ovarian and pancreatic cancers”. Similar questions are found in the text such as the sentence on page 4, lines 163-164, and page 9, lines 380-382.
  2. The mechanism about how insulin, IGF and IGFR-I contribute to the development of PDAC is well introduced. This pathway would be more complete if the mechanism about how obesity causes hyperinsulinemia could be briefly
  3. Table 1 summarized the family of FABP, the location of expression and their ligands. However, only FABP4 and FABP5 are introduced in this article and it seems that many FABP family members have little connection with PDAC. Therefore, we believe that Table 1 is unnecessary and should be removed.
  4. Figure 2 is well designed to demonstrate the mechanism introduced above. We would recommend the author to putthis picture at the beginning of the mechanism introduction. This will greatly help the readers’ 
  5. In the part of “3 Cytokines, chemokines and senescence-associated secretory phenotype (SASP)”, the authors are trying to discuss the potential relation between cellular senescence/SASP and PDAC. But we believe that the references cited in this part can’t illustrate your points. This part would be more convincing if the authors could cite more reference about the relation of PDAC and cellular senescence.

Minor Points 

  1. The article may look more concise if two citations werewritten together, as “[11,12]”.
  2. The bold fonts in Table 1vary in size, it would be much better if the size is consistent.
  3. On page 4 , lines 146-148, the authors used the words“several studies”. However, there is only one reference to prove the idea, so it would be more rigorous to use words “one study”or cite more articles. Similar questions are found in this text.
  4. On page 8, lines 352-353, the sentence should be corrected as “High fat diet (HFD) increases fat uptake in tumor cells…”

Author Response

Response to reviewer2’s comment

Thank you very much for carefully reviewing our manuscript and providing valuable comments to us. We have carefully studied the comments and have performed some experiments. We have added some more data and amended the manuscript based on them. We hope our revised manuscript meets with your approval.

Major issues1: In the introduction, page 1, lines 29-31. It would be more persuasive if articles or data are cited to prove the conclusion “obesity raises the risk of obesity related malignancy, including esophageal adenocarcinoma, endometrial, gastric, liver, renal, colorectal, breast, ovarian and pancreatic cancers”. Similar questions are found in the text such as the sentence on page 4, lines 163-164, and page 9, lines 380-382.

Answer1: We modified citations.

Major issues2: The mechanism about how insulin, IGF and IGFR-I contribute to the development of PDAC is well introduced. This pathway would be more complete if the mechanism about how obesity causes hyperinsulinemia could be briefly

Answer2: We add the sentence of “Obesity is associated with increased basal and post-prandial insulin secretion” and citation.

Major issues3: Table 1 summarized the family of FABP, the location of expression and their ligands. However, only FABP4 and FABP5 are introduced in this article and it seems that many FABP family members have little connection with PDAC. Therefore, we believe that Table 1 is unnecessary and should be removed.

Answer3: We think where FABPs express is important. To make readers easier to understand where FABPs express, we thought table1 is necessary.

Major issues4: Figure 2 is well designed to demonstrate the mechanism introduced above. We would recommend the author to put this picture at the beginning of the mechanism introduction. This will greatly help the readers’

Answer4: We put the figure at the beginning.

Major issues5: In the part of “3 Cytokines, chemokines and senescence-associated secretory phenotype (SASP)”, the authors are trying to discuss the potential relation between cellular senescence/SASP and PDAC. But we believe that the references cited in this part can’t illustrate your points. This part would be more convincing if the authors could cite more reference about the relation of PDAC and cellular senescence.

Answer5: We changed the sentence and cited more reference.

Minor issues1: The article may look more concise if two citations were written together, as “[11,12]”.

Answer1: As reviewer recommended, we modified that point.

Minor issues2: The bold fonts in Table 1vary in size, it would be much better if the size is consistent.

Answer2: We modified that point.

Minor issues3: On page 4 , lines 146-148, the authors used the words“several studies”. However, there is only one reference to prove the idea, so it would be more rigorous to use words “one study” or cite more articles. Similar questions are found in this text.

Answer3: We modified the sentence as “meta analysis has identified that VAT volume has a stronger correlation to certain gastrointestinal malignancies, including PDAC [25] given that the pancreas is anatomically surrounded by the VAT [34].”

Minor issues4: On page 8, lines 352-353, the sentence should be corrected as “High fat diet (HFD) increases fat uptake in tumor cells…”

Answer4: We corrected the sentence as you recommended.